# Butyrate and the Fine-Tuning of Colonic Homeostasis: Implication for Inflammatory Bowel Diseases

**DOI:** 10.3390/ijms22063061

**Published:** 2021-03-17

**Authors:** Naschla Gasaly, Marcela A. Hermoso, Martín Gotteland

**Affiliations:** 1Department of Nutrition, Faculty of Medicine, Universidad de Chile, Santiago 8380453, Chile; naschla.gasaly@uchile.cl; 2Laboratory of Innate Immunity, Program of Immunology, Institute of Biomedical Sciences, Faculty of Medicine, Universidad de Chile, Santiago 8380453, Chile; mhermoso@med.uchile.cl; 3Department of Human Nutrition, Institute of Nutrition and Food Technology (INTA), Universidad de Chile, Santiago 7830490, Chile; 4Millennium Nucleus in the Biology of Intestinal Microbiota, Santiago 8380453, Chile

**Keywords:** butyrate-producing bacteria, mitochondrial function, gut microbiota, stem cells, paradoxical effect, oxygen gradient, aryl hydrocarbon receptor 1, GPR41, GPR109A, HDAC

## Abstract

This review describes current evidence supporting butyrate impact in the homeostatic regulation of the digestive ecosystem in health and inflammatory bowel diseases (IBDs). Butyrate is mainly produced by bacteria from the Firmicutes phylum. It stimulates mature colonocytes and inhibits undifferentiated malignant and stem cells. Butyrate oxidation in mature colonocytes (1) produces 70–80% of their energetic requirements, (2) prevents stem cell inhibition by limiting butyrate access to crypts, and (3) consumes oxygen, generating hypoxia and maintaining luminal anaerobiosis favorable to the microbiota. Butyrate stimulates the aryl hydrocarbon receptor (AhR), the GPR41 and GPR109A receptors, and inhibits HDAC in different cell types, thus stabilizing the gut barrier function and decreasing inflammatory processes. However, some studies indicate contrary effects according to butyrate concentrations. IBD patients exhibit a lower abundance of butyrate-producing bacteria and butyrate content. Additionally, colonocyte butyrate oxidation is depressed in these subjects, lowering luminal anaerobiosis and facilitating the expansion of Enterobacteriaceae that contribute to inflammation. Accordingly, gut dysbiosis and decreased barrier function in IBD seems to be secondary to the impaired mitochondrial disturbance in colonic epithelial cells.

## 1. Introduction

The gut microbiota (GM) is currently considered as a human virtual organ. It is made up of a myriad of microorganisms which have developed a bidirectional, constant, communication with the host, partly thanks to the production of a wide range of bacterial metabolites, the most studied being the short-chain fatty acids (SCFA): Acetate, propionate, and butyrate. The epithelial cell monolayer covering the intestinal mucosa is the first interface between the microorganisms, food products, enzymes, and bile salts present in the gut lumen, and our “milieu intérieur”. Acting as a selective barrier, it allows nutrient absorption and limits the access of potential noxiae to the subjacent immune system and bloodstream [1]. This harmonious relationship between the microbiota, epithelium, and local immune system determines a healthy digestive ecosystem. However, genetic, environmental, and modern-life style factors can alter the composition and diversity of the GM (dysbiosis), the integrity of the epithelial layer (barrier function), and/or the immune system (inflammation), disrupting this homeostatic balance and favoring the development of various pathologies. Such alterations occur in metabolic diseases (obesity, type 2 diabetes, non-alcoholic steatohepatitis), immune disorders (allergy, auto-immune diseases), disturbances of the central nervous system, mood or behavior, and digestive diseases including diarrhea, constipation, irritable bowel syndrome, and inflammatory bowel diseases (IBD) [2,3,4,5], even if it remains unclear whether they are the cause or consequence of these diseases. IBD, Crohn’s disease (CD), and ulcerative colitis (UC) are chronic disorders whose incidence is currently growing world-wide. They affect mainly the gastrointestinal tract with successive periods of remission and activity, resulting in increased deterioration of the patient’s quality of life. IBDs have a multifactorial etiology, resulting from an altered homeostasis of the gastrointestinal associated lymphoid tissue (GALT) in genetically susceptible individuals under certain favorable environmental factors [6], leading to aberrant mucosa immune responses. 

In recent decades, butyrate has aroused increasing interest due to its multifaceted properties, not only in the digestive system but also in the regulation of energy metabolism. The aim of this review is to describe the current evidence supporting the different regulatory activities of butyrate involved in the homeostasis of the digestive ecosystem, both in the healthy subject and in the context of IBDs. 

## 2. The Gut Microbiota in Health and IBDs

The GM exhibits a vast inter- and intra-individual variability, being dominated by two bacterial phyla, Firmicutes and Bacteroidetes, representing more than 90% of the whole bacteria present in the gut [7]. Other sub-dominant phyla are also present, more particularly Proteobacteria, Actinobacteria, Verrucomicrobia, and Fusobacteria. Resident symbionts contribute to many physiological processes, including digestive and metabolic functions, regulation of the epithelial barrier, development, and modulation of the immune system, etc. While most intestinal microorganisms live in a mutualistic relationship with the host, some pathobionts can cause diseases under certain conditions. Diet is the factor that most impacts GM composition. The presence of high amounts of indigestible polysaccharides and polyphenols contributes to the formation of a balanced microbiota, while low-fiber, low-polyphenol, and high-fat and/or high-protein diets stimulate bacterial populations producing toxic metabolites that favor disease development [8]. Bacteroides species express a great array of glycoside hydrolases and polysaccharide lyases and act as primary degraders of complex polysaccharides, releasing simpler oligosaccharides and metabolites used by secondary fermenters, such as *Clostridium* and *Lactobacillus*. These cross-feeding mechanisms result in the formation of acetate, propionate, butyrate, and (in lower proportion) valerate [9] which was also derived, to a lesser extent, from amino acid fermentation. SCFAs have a profound impact on human health, being a colonocyte energy source, regulating glucose metabolism and the hepatic biosynthesis of triglycerides and cholesterol, inhibiting pathogen growth, and reducing intestinal inflammation, in addition to numerous systemic effects [10]. Therefore, diets high in fibers and/or polyphenols are associated with less inflammation and inflammatory diseases, due to polyphenol-derived metabolites (valerolactones, aromatic acids) and SCFAs production [11]. Consequently, IBD and colorectal cancer incidence is inversely correlated with dietary fiber intake [12,13,14,15,16]. 

Since the initial observations of Roediger’s [17] on the preferential use of butyrate by colonocytes and his hypothesis that the SCFA deficiency could result in mucosal hypoplasia and colitis, a great number of studies have evaluated the effect of mixed SCFA enema in patients with UC, CD, diversion colitis, pouchitis, radiation colitis, and infectious colitis [18]. Though some of them reported beneficial effects of SCFAs, many described contradictory results. For example, Guillemot et al. [19] did not observe any improvement of endoscopic and histologic lesions in patients with diversion colitis after 14 days of SCFA irrigation. Similarly, a recent systematic review including four studies with a total of 187 patents fails to show any benefits of SCFA enemas in IBD patients [20]. Some of the results that we present in this review will probably help explain the paradoxical results obtained from these clinical studies.

Compelling evidence from UC patients and animal models support the strong implication of GM in the initiation and maintenance of inflammatory processes [9]. The GM of IBD patients is characterized by the loss of intraindividual diversity, higher abundance of Proteobacteria, and lower of Firmicutes. Of note, the butyrate-producing bacteria *Faecalibacterium prausnitzii*, *Ruminococcus torques*, *Roseburia inulinivorans*, *Blautia faecis*, and *Clostridium lavalense* are less abundant [21,22,23,24] and consequently, luminal butyrate concentrations are lower [25] and levels of C-reactive protein are higher, reflecting a strong inflammatory status [22]. Similar changes including a high abundance of *E. coli* were observed in the mucosa-associated microbiota of CD patients, without differences between the inflamed and non-inflamed area [26,27]. Of interest, adherent-invasive *E. coli* strains were observed in more than 30% of these patients, being capable of invading the epithelium and replicating in epithelial cells and macrophages, and generating an inflammatory response characterized by an exacerbated IL1β release through NLRP3-inflammasome activation [28,29]. Alternatively, mucosa-associated fungi were also increased in CD, enriched with Basidiomycota and Ascomycota phyla, Cystofilobasidiaceae family, and *Candida glabrata* genus [30], possibly contributing to mucosal inflammation. 

Therefore, gut dysbiosis in IBD is associated with an over-activation of the immune system generating a chronic inflammatory state of the mucosa, thus affecting the epithelial barrier integrity (Figure 1) [11,31].

## 3. Butyrate Production

Butyrate represents approximately 15–23% of the total SCFAs in human stools, with concentrations varying between 10 and 25 mM [32], probably higher in the proximal colon where most dietary fiber fermentation occurs. Butyrate production depends on the amount and type of dietary fiber consumed, and the presence of butyrate-producing bacteria. Most of these bacteria belong to the Lachnospiraceae and Ruminococcaceae families from the Firmicutes phylum, but some Bacteroidetes members may also produce butyrate (Table 1) [33]. These symbionts principally synthesize butyrate from carbohydrate-derived pyruvate and, at a much lesser extent, through the lysine, glutarate, or 4-aminobutyrate pathways fueled by proteins. Around 24% of the gut bacterial community exhibits the pyruvate/acetyl CoA pathway, while the proportion of bacteria exhibiting the other pathways is lower than 8% (Table 1) [33]. Interestingly, butyrate production varies according to the fiber type and the bacterial consortium involved in its fermentation [34]. For example, Nielsen et al. [35] observed that the fermentation of arabinoxylan by *Butyrivibrio fibrisolvens* or *Eubacterium rectale* results in higher butyrate amounts than that of starch, and that with both fibers, *B. fibrisolvens* is more efficient than *E. rectale* in producing butyrate. In addition to its production by the GM, butyrate can also be provided by some foodstuffs, in particular dairy fat which are rich in SCFAs including butyrate as tributyrin (3–8% of the total fatty acids).

## 4. Butyrate Uptake by Epithelial Cells

Butyrate is a weak acid (pKa = 4.8) mainly found under its dissociated form at the physiological colonic pH (5.0–6.5). It is mostly taken by specific transporters: The proton-coupled monocarboxylate transporter-1 (MCT1, encoded by *SLC16A1* gen) and the sodium-coupled monocarboxylate transporter-1 (SMCT1, encoded by *SLC5A8* gen). MCT1 is a low affinity transporter weakly expressed in the small intestine and strongly in the proximal colon, where butyrate concentration is higher and the pH more acidic [36]. It is found in the apical membrane of the colonocyte, while its presence in the basolateral membrane is questioned [11,37]. MCT1 expression is decreased by fasting, high-protein diets, flavonoids, and caffeine, and enhanced by dietary fibers and SCFAs including butyrate, via the GPR109A receptor stimulation [38]. SMCT1 is expressed in the apical membrane, particularly in the distal ileum and colon, where it transports a variety of monocarboxylates including lactate and pyruvate. SMCT1 has a high affinity for butyrate (50 µM), functioning in the distal colon where the butyrate concentration is low. The SMCT1-mediated co-transport of butyrate/Na^+^ stimulates water reabsorption in the colon. This mechanism was used to improve oral rehydration solutions efficiency by adding fermentable fiber [39]. The presence of both transporters was also seen in the human intestinal cell lines, HT-29 and Caco-2 [40]. The intracellular butyrate concentration is also regulated by the breast cancer resistance protein (BCRP), an ATP-binding cassette (ABC) transporter (encoded by *ABCG2* gen) present in the apical cell membrane, that removes part of the butyrate from the cell into the lumen [41]. 

Butyrate oxidation decreases in experimental colitis and IBD patient’s inflamed mucosa. This is not due to a constitutive defect in the oxidative process but to a lower butyrate uptake by colonocytes caused by the downregulation of MCT1 expression in the inflamed mucosa. Such effect was confirmed in HT29 cells exposed to TNF-α and IFN-γ [42]. A low abundance of butyrate-producing bacteria and decreased butyrate availability possibly results in the same situation [17], reducing the butyrate oxidation by epithelial cells, decreasing ATP production, and finally contributing to mucosal lesion exacerbation. 

## 5. Butyrate Effects on Epithelial Cells

Part of the physiological effects of SCFAs occur through the stimulation of G-protein coupled receptors: GPR41 (free fatty acid receptor 3; FFAR3), GPR43 (free fatty acid receptor 2; FFAR2), and GPR109A (hydroxycarboxylic acid receptor 2; HCAR2). GPR43 has a higher affinity for 2–3 carbon fatty acids such as acetate and propionate, whereas GPR41 for 3–5 carbon fatty acids, notably butyrate. GPR109A, recognized as the niacin receptor, is activated by butyrate and the ketone body 3-hydroxy-butyrate. In the digestive tract, GPR41 and GPR43 are expressed in enteroendocrine, epithelial, and smooth muscle cells, as well as in enteric neurons. Additionally, GPR43 and at a lower level GPR41 are also present in immune cells including monocytes, dendritic cells, neutrophils, and eosinophils, indicating a broad role in the regulation of immune processes. Receptor activation by SCFAs trigger different signaling cascades [37]. GPR41 is coupled to Gαi and inhibits adenylyl cyclase, decreasing intracellular cAMP levels. GPR43 stimulation is linked with both Gαi and Gαq, decreasing cAMP levels and increasing cytoplasmic calcium concentrations, but also activating the β-arrestin-2 pathway, resulting in NF-κB inhibition and pro-inflammatory cytokine downregulation. GPR109A is also expressed on immune cells (macrophages and neutrophils) and its activation is coupled to the inhibitory G protein Gi/Go and recruitment of β-arrestins into the cell membrane.

By stimulating these receptors, SCFAs contribute to the immune protection of the colonic mucosa against microorganisms [43,44], hormone secretion (GLP-1, PYY, GLP-2), and regulation of colonic motility. 

## 6. Butyrate in the Colonocyte

Butyrate is oxidized to CO_2_ by the mitochondrial oxidative phosphorylation system, allowing ATP production. This phenomenon provides 70–80% of energy requirements of healthy colonocytes [45], regulating the colonic homeostasis. More particularly, it allows the colon to perform one of its main functions, the reabsorption of water and electrolytes by generating the energy necessary for Na^+^/K^+^-ATPase activity and the establishment of the sodium electrochemical gradient, favoring Na^+^ diffusion across the epithelium. Furthermore, part of the cytosolic acetyl CoA produced from butyrate is used for lipid synthesis and histone acetylation through histone acetyl transferase (HAT). When butyrate is in excess and not fully oxidized in the mitochondria, the unmetabolized part is released into the bloodstream or inhibits histone deacetylase (HDAC) activity, increasing histone acetylation and modulating colonocyte gene expression. The importance of butyrate as an energy source for colonic epithelium is illustrated in germfree mice (lacking GM and luminal butyrate) in which the expression of enzymes catalyzing the TCA cycle is reduced, resulting in a lower NADH/NAD^+^ ratio, oxidative phosphorylation, and ATP production [46]. Therefore, colonocytes are in a state of energy deprivation that stimulates AMPK activity, p27kip1 phosphorylation, and autophagy, a situation reverted by butyrate administration. A recent study also reported that the colonic epithelial mitochondria oxidize medium and long-chain fatty acids and that acyl-CoAs are reduced in colonic tissue of mice infected with *C. rodentium* (a model of colonic inflammation), in association with an altered mitochondrial number and appearance and increased fecal excretion of acyl carnitine [47]. These alterations were also described in IBD patients and in addition, a mutation of the SLC22A5 gene encoding OCTN2, a sodium-dependent L-carnitine transporter critical for fatty acid oxidation, has been proposed as a risk factor for IBD [48]. These results confirm the importance of the mitochondria in the etiopathology of these diseases. 

In addition to the production of energy, butyrate oxidation by surface epithelium colonocytes has two important effects: (1) Limiting butyrate access to stem cells located in the bottom of crypts, and (2) consuming oxygen, generating a hypoxic state preventing O_2_ diffusion to the lumen and maintaining a favorable GM anaerobic environment.

### 6.1. The Paradoxical Effect of Butyrate 

The first effect is related to the “butyrate paradox”, based on the fact that butyrate simulates the growth of healthy, mature (differentiated) colonocytes, while inhibiting that of malignant, undifferentiated cells. A recent study by Kaiko et al. [49] in primary colonic epithelial cell culture indicates that butyrate at physiological concentrations acts as a potent suppressor of stem cell proliferation by inhibiting HDAC, resulting in changes of gene expression, while mature colonocytes remained unaffected. Mature epithelial cells show a greater expression of genes involved in fatty acid oxidation (butyrate) and TCA, differing from stem cells that exhibit mainly genes involved in aerobic glycolysis (similar to cells undergoing malignant transformation, known as “Warburg effect”). Accordingly, butyrate is not efficiently oxidized in stem and cancer cells and accumulates in their cytoplasm, resulting in HDAC inhibition, increased histone acetylation and gene expression, and the subsequent attenuation of cell proliferation. In stem cells, such effect occurs by favoring the binding of Forkhead box O3 (FOXO3) transcription factor to the promoter of genes implicated in cell cycle arrest and apoptosis. This suggests that cellular differentiation determines the response to butyrate, explaining therefore the butyrate paradox. Interestingly, these authors suggest that crypt formation in the digestive epithelium could be an evolutive adaptation protecting stem cells against elevated butyrate concentrations in the lumen. To confirm their hypothesis, they used zebrafish whose intestine does not form crypts and stem cells are directly exposed to the lumen. Notably, butyrate exposure suppressed epithelial proliferation in this model, probably explaining that zebrafish GM lack butyrate-producing bacteria [49]. A similar situation exists in newborn mice and rats, whose crypts form after delivery from stem cells present in the inter-villous spaces [50]. Therefore, stem cells are exposed to the luminal content and accordingly, the luminal administration of butyrate was shown to induce mucosal injury in newborn rats [50]. On the other hand, butyrate administration reduces epithelial proliferation in crypts adjacent to ulcers, increases the ulcer size and atrophic crypt number in dextran sodium sulfate (DSS)-treated mice [49], whilst the metronidazole pre-treatment (that eliminates butyrate-producing bacteria), attenuated DSS-induced ulcerations. 

These observations have interesting implications for the human newborn. Indeed, though the intestinal crypts are formed in utero in humans, they are shallower in the newborn than in older children, with the stem cells therefore being more exposed to the luminal content. Post-natal intestinal growth is characterized by an increased stem cells number favoring crypt fission and subsequent crypt hyperplasia. These events lead to an increased crypt length (from 123 to 287 µm) [51], improving stem cell protection. Concurrently, gut colonization by butyrate-producing bacteria and changes in butyrate concentrations occur later during the first year of life. The longitudinal study of fecal microbiota and SCFAs in infants during 1 year showed the presence of butyrate in 35, 63, and 93% of them at 3, 6, and 12 months of age, respectively, with median concentrations varying from 0 to 17 and 20.7 µmol/g, respectively at these three times [52,53]. Colonization by butyrate-producing bacteria varied according to the taxa, although density did not exceed 10^7^/g during the first year of life. Based on these observations, we therefore hypothesize that gut colonization by butyrate-producing bacteria occurs later in infants in order to not interfere with stem cells during epithelium remodeling.

### 6.2. Butyrate and the Generation of Oxygen Gradient

The second effect is that butyrate β-oxidation in mature colonocytes depletes intracellular O_2_, preventing its diffusion into the lumen and maintaining the anaerobic environment necessary for strict anaerobes. This allows the establishment of a decreasing serosal to luminal O_2_ gradient (visualized with pimonidazole, an oxygen-sensible fluorochrome) contributing to GM protection [54]. At this low pO_2_ (<7.6 mm Hg, i.e., <1%O_2_), the expression of the hypoxia-inducible factor (HIF), a transcription factor regulating gut barrier function, is stabilized. Streptomycin administration in mice deplete butyrate-producing bacteria, reducing butyrate availability and colonocyte oxidation. Without butyrate, epithelial cells use glucose for energy generation, switching their metabolism from β-oxidation to anaerobic glycolysis (i.e., from high to low O_2_ consumption). A higher pO_2_ results in decreased HIF expression, eventually negatively impacting the epithelial barrier. Higher levels of O_2_ in epithelial cells are also seen in germ-free mice, due to the absence of butyrate, and in both models, butyrate administration normalized the alterations. The increased O_2_ availability associated with the streptomycin treatment and lower butyrate oxidation facilitates its diffusion to the lumen, reducing the anaerobic environment and allowing the paradoxical expansion of the aerobic pathogen *S. typhimurium* [55], with this event being prevented by butyrate administration.

Additionally, enterobacteria expansion is also favored by the presence of luminal nitrates, an important energy source for these facultative anaerobic bacteria that can use them as terminal electron acceptors for anaerobic respiration. Increased nitrate production was reported in animal models of colonic inflammation, associated with gut dysbiosis [56]. Nitrates are produced, among others, by the nitric oxide synthase 2 (NOS2) expressed in colonocytes [57]. In physiological conditions, butyrate activates the nuclear receptor peroxisome proliferator–activated receptor-γ (PPAR-γ), inhibiting NOS2 expression and lowering luminal nitrate levels. Alternatively, the streptomycin treatment, by decreasing butyrate levels, attenuates PPAR-γ activation, and enhances NOS2 expression and luminal nitrate concentrations. Increased O_2_ availability and nitrate utilization synergistically promote expansion of enterobacteria such as *E. coli* or *S. typhimurium*, leading to mucosal inflammation. Interestingly, the drug 5-ASA, largely used for IBD treatment, acts as a PPAR-γ agonist which activates the transcription of deacetylase Sirt3 involved in the regulation of mitochondrial activity. Notably, 5-ASA improved the alterations induced by antibiotics in mice, normalizing epithelial hypoxia and gut dysbiosis. 

Taken together, these results confirm the “oxygen hypothesis” proposed by Rigottier-Gois [58] suggesting that gut dysbiosis is secondary to impaired mitochondrial disturbance, but exacerbates gut inflammation.

As previously described, IBD patients exhibit deficient cellular bioenergetic with less butyrate oxidation, and gut dysbiosis characterized by Firmicutes depletion and Enterobacteriaceae enrichment. In addition, antibiotic administration, that decreases butyrate-producing bacteria, has been proposed as a risk factor for CD [59], and colonic epithelial cells from UC patients were shown to express scarce PPAR-γ compared with healthy subjects [60]. Accordingly, decreasing intestinal O_2_ could provide a novel strategy to restore microbiota and decrease inflammation in IBD patients. 

## 7. Butyrate as a Regulator of Epigenetic Processes

Depending on its concentrations and cell location, butyrate and propionate (to a lesser extent) regulate histone acetylation through two mechanisms: HDAC inhibition and HAT activation [32], [61]. By removing acetyl groups, HDAC allows histones to wrap more closely to DNA, reducing gene accessibility to the transcription factors, and thus expression. Consequently, by inhibiting HDAC or increasing histone acetylation through HAT activation, butyrate increases gene expression. When butyrate concentrations are low, it enters the mitochondria to generate acetyl-CoA, through the tricarboxylic acid cycle and the ATP-citrate lyase (ACL), stimulating histone acetylation via HAT. When concentrations are higher and exceed the TCA cycle metabolic rate, butyrate accumulates inside the nuclei and inhibits the HDAC activity, increasing histone acetylation. Although both mechanisms increase acetylation, different sets of genes are affected [61]. Generally, low butyrate doses stimulate the expression of genes implicated in cell proliferation and differentiation, while high doses inhibit them and increase apoptotic genes, therefore regulating epithelial turnover.

## 8. Anti-inflammatory Butyrate Effects 

Inflammation is a normal defense mechanism protecting the host against infection and other threats, which must be strictly regulated to avoid exacerbation leading to tissue damage and life-threatening systemic expansion. Its regulation involves negative feedback mechanisms, such as anti-inflammatory cytokine secretion, pro-inflammatory signaling inhibition, loss of inflammatory mediator receptors, and regulatory cell activation. Butyrate is a crucial, multifaceted, anti-inflammatory agent, contributing to immune tolerance, increasing intestinal T-regulatory cells (Tregs), modulating activity of macrophages [62,63], dendritic cells [64], and lymphocytes [63], and suppressing the release of pro-inflammatory cytokine (IL-17p70 and IL-23) that polarize naive CD4+ T cells toward Th1 and Th17 subtypes [65,66]. Immune-modulating effects of butyrate are exerted through HDAC inhibition or by stimulating GPR41 or GPR109A receptors present in epithelial and immune cells, possibly suppressing nuclear factor B (NF-kB) activation and upregulating PPAR-γ [67].

Several studies have determined the effects of butyrate in macrophages, monocytes, and neutrophils. GPR41 is upregulated in macrophages and monocytes exposed to LPS and its stimulation by butyrate prevents the release of nitric oxide, IL-6, and IL-12, without affecting that of TNF-α and MCP-1 [68]. Butyrate also decreases TNF-α, CINC-2αβ, and NO production by LPS-stimulated rat neutrophils, through inhibition of HDAC activity and NF-κB activation. In addition, the infiltration of rat peritoneum by neutrophils and the release of cytokines by these cells ex vivo are attenuated by tributyrin administration, through GPR41 stimulation and NF-kB inhibition [69,70]. 

In epithelial cells, the secretion and expression of IL-8 exposed to butyrate varies depending on its concentrations and the cell model used [69,71,72,73,74,75]. Butyrate (2.5–20 mM) enhanced IL-8 expression and secretion in IL1β-stimulated Caco-2 cells, while in LPS stimulated cells, IL-8 secretion only occurred after butyrate treatments. Additionally, low butyrate concentrations (0.2–1 mM) [76] inhibited IL-8 release, while higher concentrations (20–30 mM) enhanced it to the maximal level, suggesting a pro-inflammatory activity. Alternatively, other authors reported the suppression of this chemokine by butyrate in T84, HT-29, and Caco-2 cells. For example, the stimulation of Caco-2 cells with TNFα induced IL-8 and IL-6 expression through activation of NFκB p65, spleen tyrosine kinase, and mitogen-activated protein kinase pathways, and pretreatment with butyrate (0.625 mM), propionate (2.5 mM), and acetate (5 mM), suppressed this inflammatory response [67,71,72,73,75]. 

A crucial factor involved in the resolution of inflammation and healing at the mucosal interfaces is IL-22. This cytokine, released by both innate lymphoid cells (ILCs) type 3 and CD4+ T cells, acts through STAT-3 activation in gut epithelial cells, inducing antimicrobial peptide secretion and promoting the epithelial barrier function. Germ-free mice display an impaired IL-22 secretion, supporting a role for the gut microbiota in its production. SCFAs, and more particularly butyrate, were recently shown to promote IL-22 production in CD4+ T cells and ILCs by HDAC inhibition and GPR41 stimulation [77]. Butyrate stimulated IL-22 production by promoting the expression of hypoxia-inducible factor (HIF)1α and aryl hydrocarbon receptor (AhR), which are differentially regulated by Stat3 and mTOR. The improvement of intestinal inflammation by SCFA administration through IL-22 enhancement was confirmed in *C. rodentium*-infected mice. 

On the other hand, GPR41, GPR43, and GRP109A activation in immune and epithelial cells could stimulate the NOD-like receptor family, such as the pyrin domain containing 3 (NLRP3) inflammasome complex [75]. NLRP3 stimulation induces cellular events leading to the transformation of procaspase-1 to caspase-1, and IL-1β and IL-18 secretion. SCFAs, including butyrate (0.01 mM), inhibit LPS-induced autophagy and NLRP3 inflammasome activation in Caco-2 cells, as previously reported [46]. These results are interesting since AIEC strains isolated from IBD patients have been shown to activate the NLRP3 pathway in macrophages [29]. Alternatively, GPR109A stimulation also promotes an anti-inflammatory phenotype in colonic macrophages and dendritic cells, inducing Treg cell differentiation and IL-10 secretion [64] (through HDAC inhibition), thus promoting expression of retinaldehyde dehydrogenase-1 (RALDH1), an enzyme involved in the synthesis of retinoic acid and the subsequent modulation of IL-10-producing Treg cell differentiation [63,76,78]. 

In UC patients, butyrate enemas (100 mM) reduced NF-kB activation in colonic mucosa, decreasing the disease activity index and mucosal infiltration of neutrophils and lymphocytes [79]. Contrastingly, other reports showed no benefit in UC patients in remission with the same butyrate concentration [80], while enema with a combination of SCFAs (80 mM acetate, 30 mM propionate, and 40 mM butyrate enema) produced clinical remission in only a subset of these patients [81]. Although high butyrate concentrations were shown to be more inflammatory than lower in vitro, high butyrate doses used in trials might therefore have a beneficial impact. The reasons for such contradictory findings are unclear but suggest that only part of butyrate can access the colonic mucosa in these patients. 

Butyrate also affects the innate immune system by inducing expression of the antimicrobial protein, cathelicidin, in human colonic cells [82], through the participation of activator protein 1 (AP-1) and histone acetylation in the promoter region of the cathelicidin gen [83].

## 9. Butyrate as a Ligand for the Aryl Hydrocarbon Receptor (AhR)

In addition to the G protein-coupled receptor stimulation, butyrate has recently emerged as an activator of AhR, a cytoplasmic, ligand-activated, transcription factor [84]. AhR recognizes numerous xenobiotic compounds including dietary phytochemicals, food contaminants (dioxins and benzopyrene), and tryptophan-derived microbial metabolites, such as indoles. Xenobiotic binding to AhR induces its nuclear translocation where it dimerizes with the AhR nuclear translocator (ARNT), initiating gene transcription. This results in increased expression of the cytochrome P450 family 1A (CYP1A1), a phase I enzyme involved in xenobiotic conjugation and subsequent urinary elimination [85]. Notably, gut mucosa AhR activation promotes epithelial barrier homeostasis through the Notch1-dependent pathway, increases epithelial IL-10 receptor expression, and regulates immune cells (antigen presenting cells, intraepithelial lymphocytes, Th17/Th22 cells, Treg cells, and innate lymphoid cell-3; ILC3), thus demonstrating the relevant role of AhR at the interface between diet, gut microbiota, and host [83,85,86,87,88,89,90]. Accordingly, AhR knockout mice are more susceptible to DSS-induced colonic inflammation, while IBD patients exhibit a downregulated AhR activation [91,92]. More particularly, IBD patients carrying CARD9 risk alleles show a reduced production of tryptophan-derived metabolites (AhR ligands), and a higher risk of colonic inflammation [93]. 

Butyrate alters CYP1A1 and AhR expression and stimulates CYP1A1 activity by inhibiting HDAC activity in vitro and in vivo. Additionally, it decreases CYP1A1 expression and activity at low concentrations, reducing indole metabolite (such as 6-formylindolo[3,2-b]carbazole, FICZ) clearance and increasing their access to lamina propria where they enhance gut immunity by stimulating the secretion of IL-22 and IL-17 by ILC3 and Th17 cells. These cytokines increase the production of mucin and antimicrobial peptides and improve the gut barrier function. Conversely, high butyrate concentrations enhance CYP1A1 expression and activity through HDAC inhibition, promoting indole metabolite clearance, and reducing their access to the lamina propria. Therefore, butyrate contributes to commensal bacteria tolerance by increasing the expression of AhR and that of IL-10 receptor and stimulating the differentiation and expansion of gut IL-10-releasing Treg and Tr1 cells. Furthermore, butyrate at concentrations ≥ 1 mM (as well as propionate and valerate) was recently reported to directly bind AhR and activate its signaling pathway, increasing CYP1A1 expression, independently of HDAC inhibition in human intestinal cell lines [94]. These results are relevant as low colonic SCFA concentrations and butyrate-producing bacteria are seen in IBD patients. 

Therefore, butyrate attenuates inflammation through diverse mechanisms, including inhibition of HDAC and NF-kB activities, AhR, and GPR109A, GPR41 and, to a lesser extent, GPR43 receptors (Figure 2).

## 10. Butyrate and Intestinal Barrier Function

Gut barrier function is altered in many diseases including IBD [83]. Ex vivo colonic mucosa of UC patients shows low transepithelial electrical resistance (TEER) reflecting the loss of epithelial integrity and high paracellular permeability, together with the presence of erosive, ulcerative lesions and high apoptosis rate [94,95]. Additionally, a lower production of mucin 2 (MUC2), the largest component of mucin has been reported, though it is unclear whether this represents a primary defect or is secondary to the inflammation-induced epithelial damage. 

A higher urinary excretion of lactulose/mannitol, a marker of intestinal permeability has been reported in CD patients. Interestingly, gut permeability is also affected in first-degree patients’ relatives, suggesting that gut barrier disturbance is a constitutive defect and risk factor favoring disease development. Moreover, the expression of the tight-junction protein claudin-2 (involved in pore formation) is increased, while that of claudin 4 and 5 decreased in IBD patients, with these alterations being accompanied by higher plasma concentrations of zonulin, a physiological regulator of tight junction [96]. 

The butyrate impact on gut barrier function has been explored in vitro and in vivo [97]. Elevated TEER was described in Caco-2 cells exposed to butyrate 2 mM, without changes in occludin, claudin-1 and -4, and ZO-1 expression. Such effect was due to ZO-1 and occludin redistribution through AMPK activation, favoring tight junction assembly [97]. Alternatively, exposure to higher butyrate concentrations (8 mM) results in decreased TEER and increased paracellular inulin permeability, probably due to cell viability loss and enhanced apoptosis rate [98]. Additionally, butyrate reverses LPS-induced permeability alterations (TEER and dextran-FITC permeability) by increasing claudins-3 and 4 expression in IPEC-J2 intestinal cells. Furthermore, butyrate prevents LPS-induced downregulation of Akt and 4E-BP1 phosphorylation, possibly enhancing tight junction protein abundance through Akt/mTOR-mediated protein synthesis. Low concentrations of butyrate (0.1–5 mM) improved TEER and dextran-FITC permeability and increased MUC2 and MCT expression in the human colonic cell line HT29-MTX-E12 (displaying a goblet-like cell phenotype) exposed to the mycotoxin deoxynivalenole [35]. Butyrate at supraphysiological concentrations (50–100 mM) inhibits the expression of MUC2 and MUC5AC whilst increasing that of caspase 3, catalase, and superoxide dismutase 2, suggesting higher oxidative stress and apoptosis rate [35]. Part of these results were confirmed by Finnie et al. who report enhanced MUC2 production in colonic biopsies from UC patients exposed to butyrate 0.5 mM [99]. However, [80] butyrate (100 mM) enema does not affect the expression of MUC2 and intestinal trefoil factor (ITF, a cysteine-rich, trefoil, peptide produced by goblet cells and involved in ulcer healing), sialomucin proportion, mucus production, and sIgA concentrations in UC patients and healthy subjects. In a murine model of inflammation induced by *Citrobacter rodentium*, butyrate protects the barrier function, increased expression of ITF and RELM-β (another goblet cell peptide in mucosal protection), and reversed the inflammation [100]. Additionally, butyrate administration in drinking water improved mucosal inflammation and gut barrier dysfunction in TNBS-induced experimental colitis in mice, through GPR109A stimulation and inhibition of AKT and NF-κB p65 signaling pathways [101].

Antibacterial peptides are also implicated in gut mucosal protection. Defensin- and lysozyme-producing Paneth cells are dysfunctional in CD patients, facilitating dysbiosis in mucosa-associated microbiota and increasing susceptibility to intestinal inflammation. Interestingly, Paneth cells express GPR41, GPR43, and GPR109A receptors and enhance α-defensin secretion and the subsequent of *S. thyphymurium* death when stimulated by butyrate [102]. Additionally, piglet infected with *E. coli* O157:H7 and supplemented with sodium butyrate (0.2% *W*/*W*) increased defensin expression via HDAC inhibition, improving disease resistance, alleviating symptoms, inflammation, and promoting pathogen clearance [103]. 

## 11. Role of Butyrate in the Development of Colorectal Cancer

Patients with UC or CD have a higher risk of developing colorectal cancer (CRC), and IBD-associated CRC has worse prognosis than sporadic CRC. Though genetic, environmental, dietary, and microbial factors have been involved in CRC development, the pathological mechanisms contributing to the transition from IBD to CRC remain unclear. The importance of microbiota in CRC development is supported by the fact that the administration of stool samples from CRC patients to germ-free and conventional mice treated with the carcinogen azoxymethane increased polyp number, intestinal dysplasia and proliferation, inflammation markers, and the proportions of colonic Th1 and Th17 cells, compared with the animals treated with fecal samples from healthy individuals [104]. Furthermore, several pathobionts including strains of *Bacteroides fragilis* and *E. coli* producing colibactin and fragilysin toxins, respectively, and *Fusobacterium nucleatum* expressing FasA and Fap2 adhesins have been involved in colonic inflammation and malignant transformation [105]. 

The role of butyrate in CRC development is controversial. A lower abundance of butyrate-producing bacteria has been reiteratively described in several studies in CRC patients, and the administration of *Butyricicoccus pullicaecorum* to dimethylhydrazine-treated mice improved the clinical outcome of CRC through the activation of GPR45 and MCT1 expression [106]. As previously reported, butyrate is also linked to the prevention of CRC due to its inhibitory activity of histone deacetylase (HDACi) and the promotion of cell cycle arrest, differentiation, and/or apoptosis of colorectal cancer cells at physiological concentrations [107], an effect associated with less expression of p21. Zagato et al. [108] characterized two antitumorigenic bacteria, *Faecalibaculum rodentium* in murine GM and its human homologue *Holdemanella biformis* (from the Erysipelotrichaceae family) whose abundance was impoverished in the early stages of tumorigenesis. These bacteria can inhibit cell proliferation through SCFA production and the subsequent inhibition of calcineurin and NFATc3 activation in murine and human settings. *H. biformis* was reduced in the stool of patients with large adenomas and could be used as a potential target to attenuate tumorigenesis [108]. Lower colonic butyrate levels or molecular deletion of GPR109A were associated to decrease Treg differentiation, resulting in increased polyp formation in Apc mice. Accordingly, these findings support a tumor-suppressive role for butyrate. 

In opposition with these results, Belcheva et al. [109], using a model of human adenomatous polyposis, the Apc ^Min/+^ mice also deficient for the DNA mismatch repair gene MutS homolog 2 (MSH2^−/−^), showed that the antibiotic treatment or low-carbohydrate diet decreased colonic and intestinal polyp formation by 6-fold and 2-fold, respectively. This phenomenon was associated with less abundance of butyrate producing Clostridiaceae, Lachnospiraceae, and Ruminococcaceae and less butyrate concentrations. Confirming these results, butyrate supplementation reversed polyp diminution, decreasing p21 expression and increasing epithelial cell proliferation and tumor progression, suggesting that this SCFA could act as an oncometabolite. 

Though the explanation for these paradoxical effects of butyrate on the CRC issue are unclear, it could be due to the different genetic backgrounds of the animals used in these studies [110]. 

## 12. Conclusions and Perspectives

Butyrate is a crucial link between the GM, colonic epithelium, and local immune system, with varying concentrations according to indigestible polysaccharides in the diet, and low butyrate concentrations likely contributing to IBD development or exacerbation. Butyrate attenuates inflammation through diverse mechanisms, including inhibition of HDAC and NF-kB activities, and stimulation of PPAR-γ, AhR, and GPR109A, GPR41 and, to a lesser extent, GPR43 receptors. An important feature of butyrate is its oxidation in the colonocyte and the subsequent generation of an O_2_ gradient contributing to colonic lumen anaerobiosis. This phenomenon allows maintaining a healthy microbiota in the colon, avoiding overgrowth of potentially pro-inflammatory enterobacteria. Decreased butyrate-producing bacteria and butyrate concentrations in IBD patients affect the colonocyte energy metabolism and inflammation development. Hence, maintaining adequate colonic butyrate levels is a promising strategy for IBD patients. Though direct oral butyrate administration is limited by its low palatability, the use of tributyrin, a triglyceride naturally present in dairy fat, is more acceptable. On the other hand, butyrate and tributyrin are rapidly absorbed in the small intestine [111], with probably few molecules reaching the colon. These limitations could be circumvented using micro- or nano-encapsulated butyrate. Alternatively, butyrate-releasing derivates such as 4-phenylbutyrate, butyrate acyloxy alkyl ester, and N-(1-carbamoyl-2-phenylethyl) butyramide have been recently developed, and their administration in animal models of colonic inflammation improved GM and inflammation [112]. Additionally, butyrate formation in the colon can be induced through the administration of dietary fibers or prebiotics, although these differ in their ability to produce butyrate through fermentation. 

Finally, the use of butyrate-producing bacteria as probiotics or biotherapeutic agents is interesting although with several limitations, as these bacteria are strict anaerobic and hard to cultivate, limiting the obtention of sufficient biomass for a commercial goal. In addition, whether GRAS culture media are currently available is unclear and their safety must be closely confirmed.

## Figures and Tables

**Figure 1 ijms-22-03061-f001:**
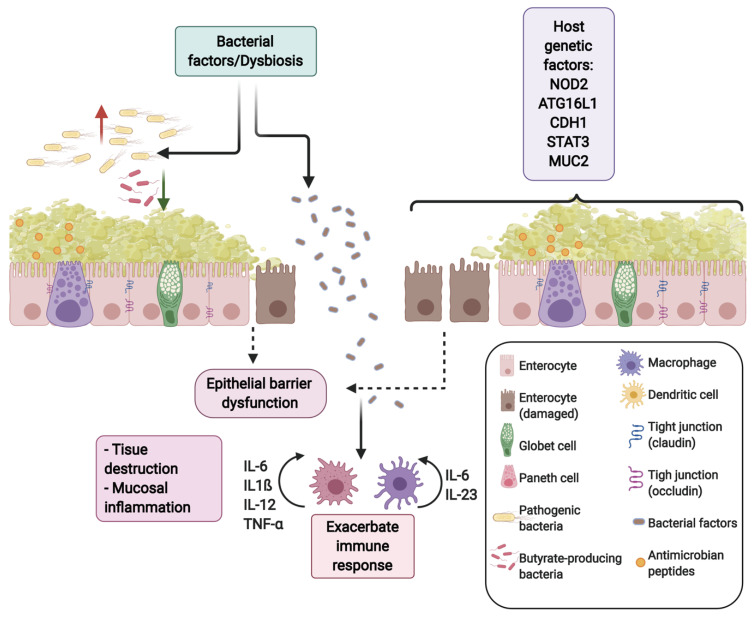
Genetic factors, alterations in the intestinal microbiota, and an exacerbated immune response cause a defect in the intestinal barrier function, affecting its integrity, increasing tissue destruction, and mucosal inflammation.

**Figure 2 ijms-22-03061-f002:**
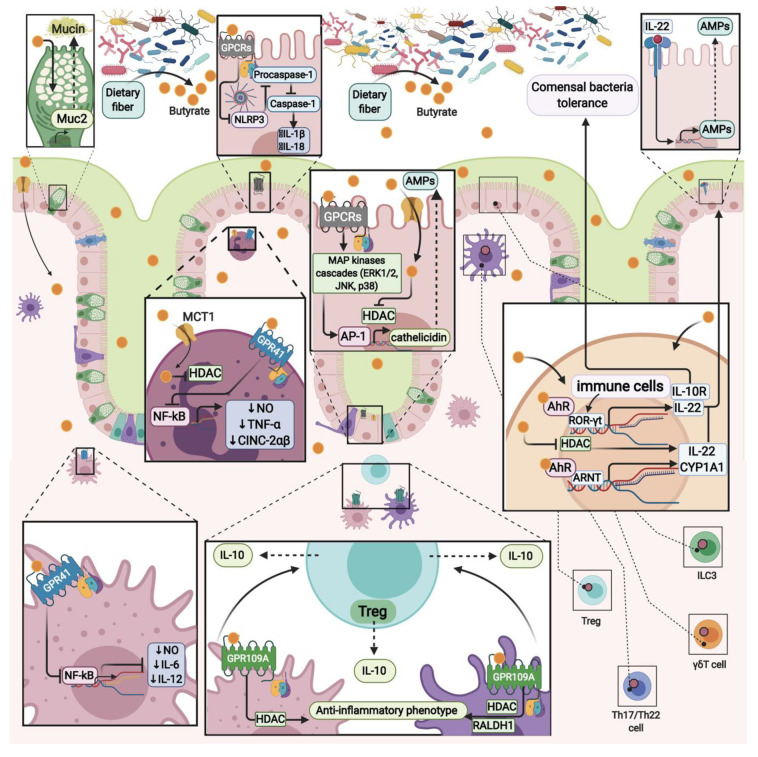
Different anti-inflammatory mechanisms of butyrate: histone deacetylase (HDAC) inhibition, aryl hydrocarbon receptor (AhR) and G-protein coupled receptors (GPCRs) activation.

**Table 1 ijms-22-03061-t001:** Classification of the main butyrate-producing bacteria from the Firmicutes and Bacteroidetes phyla according to the different pathways they use to synthesize butyrate (from [33]).

Butyrate Pathway	Firmicutes		Bacteroidetes	GM Abundance of the Bacterial Populations Expressing Each Pathway
	Lachnospiraceae	Rumicococcaceae		
Acetyl-CoA	*Eubacterium* *Butyrivibrio* *Clostridium XIVa* *Ruminococcus* *Anaerostipes* *Coprococcus*	*Subdoligranulum* *Butyricicoccus* *Roseburia* *Pseudoflavonifractor* *Flavonifractor* *Oscillibacter* *Faecalibacterium*	*Odoribacter* *Butyricimonas* *Porphyromonas*	24.2%
Glutarate	*Clostridium XIVa*	*Pseudoflavonifractor* *Oscillibacter*		1.8%
4-aminobutyrate		*Flavonifractor*	*Odoribacter* *Butyricimonas*	1.7%
Lysine		*Flavonifractor*	*Odoribacter* *Alistipes* *Butyricimonas*	4.4%

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
