# Peer review of "Butyrate and the Fine-Tuning of Colonic Homeostasis: Implication for Inflammatory Bowel Diseases"

_ijms, 2021, doi:10.3390/ijms22063061_

Round 1
Reviewer 1 Report
The paper "Butyrate and the fine-tuning of colonic homeostasis" presented to me for evaluation. Implication for inflammatory bowel diseases" deals with a scientifically interesting topic. The paper contains 13 pages of text. The paper includes 100 current literature references. Moreover, the authors have included 2 figures and 1 table.
The paper presents in a modern and detailed way the role of butyrate in homeostasis of bacterial flora of the large bowel.
In Chapter 2 the authors presented in an interesting and detailed way the current state of knowledge about intestinal microbiota in both healthy individuals and IBD cases. In the further parts of the paper the authors discussed all aspects of butyrate function in colonic metabolism. In Figure 2 they presented a scheme of anti-inflammatory action of butyrate. In Chapter 11 the authors briefly summarize the whole publication. I believe that the work presented to me for review is valuable. It presents the current state of knowledge on the discussed subject.
In my opinion the paper can be accepted for publishing in its current form.
Author Response
Thank you very much for your comments. It is very pleasing to us that the work carried out has your full approval.
Reviewer 2 Report
Interesting review on the role of short chain fatty acid in the colonic homeostasis. The authors focused in their review on the physiological role of SCFA in the gut. Especially the antiinflammatory effect of butyrate is extensively described.
Points of criticism:
- I miss in this paper the link between SCFA production by microbiota and the clinic. What is the importance of SCFA in the prevention of diseases such as IBD or cancer ?
- The role of SCFA in the prevention of colorectal cancer and the antineoplastic effects of SCFA should be added
- SCFA stimulate the production of IL22 in the enterocytes and this mechanism plays an important antiinflammatory role. The authors should discuss this aspect
- There is an evidence that inflammatory states in the gut lead to the mitochondrial dysfunction and reduced butyrate oxygenation. Please include this aspect in this review (Smith et al, JCI 2020)
Author Response
We thank the reviewer for his helpful comments. All changes made in the text are highlighted in yellow.
1. I miss in this paper the link between SCFA production by microbiota and the clinic. What is the importance of SCFA in the prevention of diseases such as IBD or cancer?
A paragraph about the health impact of SCFAs in IBD patients was added L83-91. Regarding Colorectal cancer, new data supporting the controversial role o butyrate in this disease was incorporated L452-467
2. The role of SCFA in the prevention of colorectal cancer and the antineoplastic effects of SCFA should be added
According to the reviewer’s comment, a paragraph (N”11) on the Role of butyrate in the development of colorectal cancer was included in the text of the manuscript, L441-477.
3. SCFA stimulate the production of IL22 in the enterocytes and this mechanism plays an important anti-inflammatory role. The authors should discuss this aspect
We agree with the reviewer about the crucial role of IL22 in gut homeostasis. A paragraph about this matter was therefore incorporated in the text (L326-335).
4. There is an evidence that inflammatory states in the gut lead to the mitochondrial dysfunction and reduced butyrate oxygenation. Please include this aspect in this review (Smith et al, JCI 2020)
The importance of butyrate in the generation of oxygen gradient is described in the paragraph 6.2 (L245-282). According to the reviewer’s comment, we also added a paragraph describing the findings reported by Smith el al (L192-199)